# MULTI-AGENT QUERY REFORMULATION: CHALLENGES AND THE ROLE OF DIVERSITY

**Rodrigo Nogueira**[*]
New York University
`rodrigonogueira@nyu.edu`

**Jannis Bulian, Massimiliano Ciaramita**
Google AI Language
`{jbulian,massi}@google.com`

## ABSTRACT

We investigate methods to efficiently learn diverse strategies in reinforcement learning for a generative structured prediction problem: query reformulation. In the proposed framework an agent consists of multiple specialized sub-agents and a meta-agent that learns to aggregate the answers from sub-agents to produce a final answer. Sub-agents are trained on disjoint partitions of the training data, while the meta-agent is trained on the full training set. Our method makes learning faster, because it is highly parallelizable, and has better generalization performance than strong baselines, such as an ensemble of agents trained on the full data. We evaluate on the tasks of document retrieval and question answering. The improved performance seems due to the increased diversity of reformulation strategies. This suggests that multi-agent, hierarchical approaches might play an important role in structured prediction tasks of this kind. However, we also find that it is not obvious how to characterize diversity in this context, and a first attempt based on clustering did not produce good results. Furthermore, reinforcement learning for the reformulation task is hard in high-performance regimes. At best, it only marginally improves over the state of the art, which highlights the complexity of training models in this framework for end-to-end language understanding problems.

## 1 INTRODUCTION

Reinforcement learning (RL) has proven effective in several language tasks, such as machine translation (Wu et al., 2016; Ranzato et al., 2015; Bahdanau et al., 2016), question-answering (Wang et al., 2017a; Hu et al., 2017), and text summarization (Paulus et al., 2017). In RL efficient exploration is key to achieve good performance. The ability to explore in parallel a diverse set of strategies often speeds up training and leads to a better policy (Mnih et al., 2016; Osband et al., 2016).

In this work, we propose a simple method to achieve efficient parallelized exploration of diverse policies, inspired by hierarchical reinforcement learning (Singh, 1992; Lin, 1993; Dietterich, 2000; Dayan & Hinton, 1993). We structure the agent into multiple *sub-agents*, which are trained on disjoint subsets of the training data. Sub-agents are co-ordinated by a meta-agent, called *aggregator*, that groups and scores answers from the sub-agents for each given input. Unlike sub-agents, the aggregator is a generalist since it learns a policy for the entire training set. We argue that it is easier to train multiple sub-agents than a single generalist one since each sub-agent only needs to learn a policy that performs well for a subset of examples. Moreover, specializing agents on different partitions of the data encourages them to learn distinct policies, thus giving the aggregator the possibility to see answers from a population of diverse agents. Learning a single policy that results in an equally diverse strategy is more challenging. Since each sub-agent is trained on a fraction of the data, and there is no communication between them, training can be done faster than training a single agent on the full data. Additionally, it is easier to parallelize than applying existing

---

[*]Work done while interning at Google.

distributed algorithms such as asynchronous SGD or A3C (Mnih et al., 2016), as the sub-agents do not need to exchange weights or gradients. After training the sub-agents, only their actions need to be sent to the aggregator. We build upon the works of Nogueira & Cho (2017) and Buck et al. (2018b). Hence, we evaluate our method on the same tasks: query reformulation for document retrieval and question-answering. We show that it outperforms a strong baseline of an ensemble of agents trained on the full dataset. We also found that performance and reformulation diversity are correlated (Sec. 5.5). Our main contributions are the following:

- A simple method to achieve more diverse strategies and better generalization performance than a model average ensemble.
- Training can be easily parallelized in the proposed method.
- An interesting finding that contradicts our, perhaps naive, intuition: specializing agents on semantically similar data does not work as well as random partitioning. An explanation is given in Appendix F.
- We report new state-of-the art results on several datasets using BERT (Devlin et al., 2018). However results improve marginally using reinforcement learning and on the question answering task we see no improvements.

## 2 RELATED WORK

The proposed approach is inspired by the mixture of experts, which was introduced more than two decades ago (Jacobs et al., 1991; Jordan & Jacobs, 1994) and has been a topic of intense study since then. The idea consists of training a set of agents, each specializing in some task or data. One or more gating mechanisms then select subsets of the agents that will handle a new input. Recently, Shazeer et al. (2017) revisited the idea and showed strong performances in the supervised learning tasks of language modeling and machine translation. Their method requires that output vectors of experts are exchanged between machines. Since these vectors can be large, the network bandwidth becomes a bottleneck. They used a variety of techniques to mitigate this problem. Anil et al. (2018) later proposed a method to further reduce communication overhead by only exchanging the probability distributions of the different agents. Our method, instead, requires only scalars (rewards) and short strings (original query, reformulations, and answers) to be exchanged. Therefore, the communication overhead is small.

Previous works used specialized agents to improve exploration in RL (Dayan & Hinton, 1993; Singh, 1992; Kaelbling et al., 1996). For instance, Stanton & Clune (2016) and Conti et al. (2017) use a population of agents to achieve a high diversity of strategies that leads to better generalization performance and faster convergence. Rusu et al. (2015) use experts to learn subtasks and later merge them into a single agent using distillation (Hinton et al., 2015). The experiments are often carried out in simulated environments, such as robot control (Brockman et al., 2016) and video-games (Bellemare et al., 2013). In these environments, rewards are frequently available, the states have low diversity (e.g., same image background), and responses usually are fast (60 frames per second). We, instead, evaluate our approach on tasks whose inputs (queries) and states (documents and answers) are diverse because they are in natural language, and the environment responses are slow (0.5-5 seconds per query).

Somewhat similarly motivated is the work of Serban et al. (2017). They train many heterogeneous response models and further train an RL agent to pick one response per utterance.

## 3 METHOD

### 3.1 TASK

We describe our setup using a generic end-to-end search task. The problem consists in learning to reformulate a query so that the underlying retrieval system can return a better result.

We frame the problem as an RL task (Nogueira & Cho, 2017; Buck et al., 2018b), in which the query reformulation system is an RL-agent that interacts with an environment that provides answers and rewards. The goal of the agent is to generate reformulations such that the expected returned reward

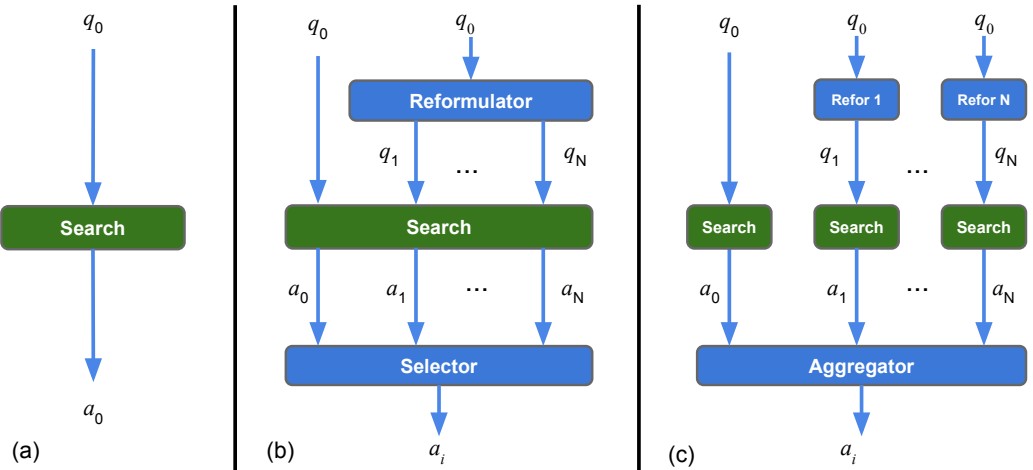

Figure 1: **a)** A vanilla search system. The query $q_0$ is given to the system which outputs a result $a_0$. **b)** The search system with a reformulator. The reformulator queries the system with $q_0$ and its reformulations $\{q_1, ...q_N\}$ and receives back the results $\{a_0, ..., a_N\}$. A selector then decides the best result $a_i$ for $q_0$. **c)** The proposed system. The original query is reformulated multiple times by different reformulators. Reformulations are used to obtain results from the search system, which are then sent to the aggregator, which picks the best result for the original query based on a learned weighted majority voting scheme. Reformulators are independently trained on disjoint partitions of the dataset thus increasing the variability of reformulations.

(i.e., correct answers) is maximized. The environment is treated as a black-box, i.e., the agent does not have direct access to any of its internal mechanisms. Figure 1-(b) illustrates this framework.

## 3.2   SYSTEM

Figure 1-(c) illustrates the agent. An input query $q_0$ is given to the $N$ sub-agents. A sub-agent is any system that accepts as input a query and returns a corresponding reformulation. Thus, sub-agents can be heterogeneous. Here we train each sub-agent on a partition of the training set. The $i$-th agent queries the underlying search system with the reformulation $q_i$ and receives a result $a_i$. The set $\{(q_i, a_i)|0 \le i \le N\}$ is given to the aggregator, which then decides which result will be final.

## 3.3   SUB-AGENTS

The first step for training the agent is to partition the training set. We randomly split it into equal-sized subsets. For an analysis of how other partitioning methods affect performance, see Appendix F. In our implementation, a sub-agent is a sequence-to-sequence model (Sutskever et al., 2014; Cho et al., 2014) trained on a partition of the dataset. It receives as an input the original query $q_0$ and outputs a list of reformulated queries $(q_i)$ using beam search. Each reformulation $q_i$ is given to the same environment that returns a list of results $(a_i^1, .., a_i^K)$ and their respective rewards $(r_i^1, ..r_i^K)$. We then use REINFORCE (Williams, 1992) to train the sub-agent. At training time, instead of using beam search, we sample reformulations. We also add the identity agent (i.e., the reformulation is the original query) to the pool of sub-agents.

## 3.4   META-AGENT: AGGREGATOR

The aggregator receives as inputs $q_0$ and a list of candidate results $(a_i^1, ..a_i^K)$ for each reformulation $q_i$. We first compute the set of unique results $a_j$ and two different scores for each result: the accumulated rank score $s_j^A$ and the relevance score $s_j^R$.

The accumulated rank score is computed as $s_j^A = \sum_{i=1}^{N} \frac{1}{\text{rank}_{i,j}}$, where $\text{rank}_{i,j}$ is the rank of the j-th result when retrieved using $q_i$. The relevance score $s_j^R$ is the prediction that the result $a_j$ is relevant

to query $q_0$. It is computed as:

$$s_j^R = \sigma(W_2 \text{ReLU}(W_1 z_j + b_1) + b_2),$$ (1)

where

$$z_j = [f_{\text{CNN}}(q_0); f_{\text{BOW}}(a_j); f_{\text{CNN}}(q_0) - f_{\text{BOW}}(a_j); f_{\text{CNN}}(q_0) \odot f_{\text{BOW}}(a_j)],$$ (2)

$W_1 \in \mathbb{R}^{4D \times D}$ and $W_2 \in \mathbb{R}^{D \times 1}$ are weight matrices, $b_1 \in \mathbb{R}^D$ and $b_2 \in \mathbb{R}^1$ are biases. The brackets in $[x; y]$ represent the concatenation of vectors $x$ and $y$. The symbol $\odot$ denotes the element-wise multiplication, $\sigma$ is the sigmoid function, and ReLU is a Rectified Linear Unit function (Nair & Hinton, 2010). The function $f_{\text{CNN}}$ is implemented as a CNN encoder[1] followed by average pooling over the sequence (Kim, 2014). The function $f_{\text{BOW}}$ is the average word embeddings of the result. At test time, the top-K answers with respect to $s_j = s_j^A s_j^R$ are returned.

We train the aggregator with stochastic gradient descent (SGD) to minimize the cross-entropy loss:

$$L = -\sum_{j \in J^*} \log(s_j^R) - \sum_{j \notin J^*} \log(1 - s_j^R),$$ (3)

where $J^*$ is the set of indexes of the ground-truth results. The architecture details and hyperparameters can be found in Appendix B.

## 4 DOCUMENT RETRIEVAL

This task involves rewriting a query to improve the relevance of a search engine's results.

### 4.1 ENVIRONMENT

The environment receives a query and returns a list of documents, the observation, and a reward computed using a list of ground truth documents. We use Lucene[2] in its default configuration as our search engine, with BM25 ranking. The input is a query and the output is a ranked list of documents.

### 4.2 DATASETS

**TREC-CAR:** Introduced by Dietz et al. (2017), in this dataset the input query is the concatenation of a Wikipedia article title with the title of one of its section. The ground-truth documents are the paragraphs within that section. The corpus consists of all of the English Wikipedia paragraphs, except the abstracts. The released dataset has five predefined folds, and we use the first four as a training set (approx. 3M queries), and the remaining as a validation set (approx. 700k queries). The test set is the same used evaluate the submissions to TREC-CAR 2017 (approx. 1,800 queries).

**Jeopardy:** This dataset was introduced by Nogueira & Cho (2016). The input is a *Jeopardy!* question. The ground-truth document is a Wikipedia article whose title is the answer to the question. The corpus consists of all English Wikipedia articles.

**MSA:** Introduced by Nogueira & Cho (2017), this dataset consists of academic papers crawled from Microsoft Academic API.[3] A query is the title of a paper and the ground-truth answer consists of the papers cited within. Each document in the corpus consists of its title and abstract.

### 4.3 REWARD

The goal of query reformulation is to increase the proportion of relevant documents returned. We use recall as the reward: $\text{R@}K = \frac{|D_K \cap D^*|}{|D^*|}$, where $D_K$ are the top-$K$ retrieved documents and $D^*$ are the relevant documents. We also experimented using other metrics such as NDCG, MAP, MRR, and R-Precision but these resulted in similar or slightly worse performance than Recall@40. Despite the agents optimizing for Recall, we report the main results in MAP as this is a more commonly used metric in information retrieval. For results in other metrics, see Appendix A.

---

[1]In the preliminary experiments, we found CNNs to work better than LSTMs (Hochreiter & Schmidhuber, 1997).

[2]https://lucene.apache.org/

[3]https://www.microsoft.com/cognitive-services/en-us/academic-knowledge-api

| | TREC-CAR | Jeopardy | MSA | Training (Days) | FLOPs ($\times 10^{18}$) |
|---|---|---|---|---|---|
| BM25 | 11.3 | 8.2 | 3.1 | N/A | |
| PRF | 11.6 | 13.1 | 3.4 | N/A | |
| RM3 | 12.0 | 13.5 | 3.1 | N/A | |
| RL-RNN (Nogueira & Cho, 2017) | 12.8 | 15.9 | 4.1 | 10 | 2.3 |
| RL-10-Ensemble | 13.0 | 17.0 | 4.4 | 10 | 23.0 |
| RL-10-Full | 14.1 | 29.3 | 4.9 | 1 | 2.3 |
| RL-10-Bagging | 14.1 | 29.6 | 5.0 | 1 | 2.3 |
| RL-10-Sub | 14.3 | 30.5 | 5.5 | 1 | 2.3 |
| RL-10-Sub (Pretrained) | 14.4 | 30.7 | 5.4 | 10*+1 | 4.6 |
| RL-10-Full (Extra Budget) | 14.8 | 31.2 | 5.6 | 10 | 23.0 |
| RL-10-Full (Ensemble 10 Aggregators) | 17.7 | 33.9 | 6.1 | 10 | 23.0 |
| RM3 + BERT Aggregator | 35.5 | 41.3 | 6.6 | 10 | 23.0 |
| RL-10-Sub + BERT Aggregator | 36.4 | 42.5 | 7.2 | 10 | 23.0 |
| Best System of TREC-CAR 2017 (MacAvaney et al., 2017) | 14.8 | - | - | - | - |

Table 1: MAP scores on the test sets of the document retrieval datasets. *The weights of the agents are initialized from a single model pretrained for ten days on the full training set.

## 4.4 BASELINES

**BM25:** We give the original query to Lucene with BM25 as a ranking function and use the retrieved documents as results.

**PRF:** This is the pseudo relevance feedback method (Rocchio, 1971). We expand the original query with terms from the documents retrieved by the Lucene search engine using the original query. The top-N TF-IDF terms from each of the top-K retrieved documents are added to the original query, where N and K are selected by a grid search on the validation data.

**Relevance Model (RM3):** A re-implementation of the query expansion model of Lavrenko & Croft (2001). The probability of adding a term $t$ to the original query is given by:

$$P(t|q_0) = (1 - \lambda)P'(t|q_0) + \lambda \sum_{d \in D_0} P(d)P(t|d)P(q_0|d), \tag{4}$$

where $P(d)$ is the probability of retrieving the document $d$, assumed uniform over the set, $P(t|d)$ and $P(q_0|d)$ are the probabilities assigned by the language model obtained from $d$ to $t$ and $q_0$, respectively. $P'(t|q_0) = \frac{\text{tf}(t \in q)}{|q|}$, where $\text{tf}(t, d)$ is the term frequency of $t$ in $d$. We set the interpolation parameter $\lambda$ to 0.65, which was the best value found by a grid-search on the development set.

We use a Dirichlet smoothed language model (Zhai & Lafferty, 2001) to compute a language model from a document $d \in D_0$:

$$P(t|d) = \frac{\text{tf}(t, d) + uP(t|C)}{|d| + u}, \tag{5}$$

where $u$ is a scalar constant ($u = 1500$ in our experiments), and $P(t|C)$ is the probability of $t$ occurring in the entire corpus $C$.

We use the $N$ terms with the highest $P(t|q_0)$ in an expanded query, where $N = 100$ was the best value found by a grid-search on the development set.

**RL-RNN:** This is the sequence-to-sequence model trained with reinforcement learning from Nogueira & Cho (2017). The reformulated query is formed by appending new terms to the original query. The terms are selected from the documents retrieved using the original query. The agent is trained from scratch.

**RL-N-Ensemble:** We train $N$ RL-RNN agents with different initial weights on the full training set. At test time, we average the probability distributions of all the $N$ agents at each time step and select the token with the highest probability, as done by Sutskever et al. (2014).

## 4.5 PROPOSED MODELS

We evaluate the following variants of the proposed method:

**RL-N-Full:** We train $N$ RL-RNN agents with different initial weights on the full training set. The answers are obtained using the best (greedy) reformulations of all the agents and are given to the aggregator.

**RL-N-Bagging:** This is the same as RL-N-Full but we construct the training set of each RL-RNN agent by sampling with replacement D times from the full training set, which has a size of D. This is known as the bootstrap sample and leads to approximately 63% unique samples, the rest being duplicates.

**RL-N-Sub:** This is the proposed agent. It is similar to RL-N-Full but the multiple sub-agents are trained on random partitions of the dataset (see Figure 1-(c)).

**BERT Aggregator:** We experimented replacing our aggregator with BERT (Devlin et al., 2018), which holds state-of-the-art results in a wide range of textual tasks. Using the same notation used in their paper, we feed the query as sentence A and the document text as sentence B. We truncate the document text such that concatenation of query, document, and separator tokens have a maximum length of 512 tokens. We use a pretrained BERT$_{\text{LARGE}}$ model as a binary classification model, that is, we feed the [CLS] vector to a single layer neural network and obtain the probability of the document being correct. We obtain the final list of documents by ranking them with respect these probabilities. We train with the same objective used to train our aggregator (Equation 3). To compare how well our proposed reformulation agents perform against the best non-neural reformulation method, we implemented two variants of the system. In one the initial list of candidate documents $a_j$ is given by RM3 (RM3 + BERT Aggregator), in the other by RL-10-Sub (RL-10-Sub + BERT Aggregator).

## 4.6 RESULTS

A summary of the document retrieval results is shown in Table 1. We estimate the number of floating point operations used to train a model by multiplying the training time, the number of GPUs used, and 2.7 TFLOPS as an estimate of the single-precision floating-point of a K80 GPU. Since the sub-agents are frozen during the training of the aggregator, we pre-compute all $(q_0, q_i, a_i, r_i)$ tuples from the training set, thus avoiding sub-agent or environment calls. This reduces its training time to less than 6 hours ($0.06 \times 10^{18}$ FLOPs). Since this cost is negligible when compared to the sub-agents', we do not include it in the table.

The proposed methods (RL-10-{Sub, Bagging, Full}) have 20-60% relative performance improvement over the standard ensemble (RL-10-Ensemble) while training ten times faster. More interestingly, RL-10-Sub has a better performance than the single-agent version (RL-RNN), uses the same computational budget, and trains on a fraction of the time. Lastly, we found that RL-10-Sub (Pre-trained) has the best balance between performance and training cost across all datasets. Compared to the top-performing system in the TREC-CAR 2017 Track (MacAvaney et al., 2017), an RL-10-Full with an ensemble of 10 aggregators yields a relative performance improvement of approximately 20%. By replacing our aggregator with BERT, we improve performance by 50-100% in all three datasets (RL-10-Sub + BERT Aggregator). This is a remarkable improvement given that we used BERT without any modification from its original implementation. Without using our reformulation agents, the performance drops by 3-10% (RM3 + BERT Aggregator). For an analysis of the aggregator's contribution to the overall performance, see Appendix C.

We compare performance of the full system (reformulators + aggregator) for different numbers of agents in Figure 2. The performance is stable across all datasets after more than ten sub-agents are used, indicating robustness. For more related experiments, see Appendix D.

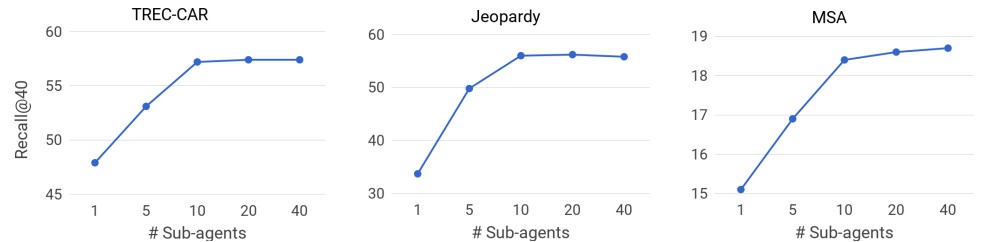

Figure 2: Overall system's performance for different number of sub-agents.

| | Dev | | Test | | Training | FLOPs |
| | F1 | Oracle | F1 | Oracle | (Days) | ($\times 10^{18}$) |
|---|---|---|---|---|---|---|
| BiDAF (Seo et al., 2016) | 37.9 | - | 34.6 | - | N/A | |
| R$^3$ (Wang et al., 2017a) | - | - | 55.3 | - | N/A | |
| Re-Ranker (Wang et al., 2017b) | - | - | 60.6 | - | N/A | |
| DS-QA (Lin et al., 2018) | - | - | 64.5 | - | N/A | |
| BERT (our run) | 71.2 | - | 69.1 | - | N/A | |
| AQA (Buck et al., 2018b) | 47.4 | 56.0 | 45.6 | 53.8 | 10 | 4.6 |
| BiDAF + AQA-10-Sub | 51.7 | 66.8 | 49.0 | 61.5 | 1 | 4.6 |
| BiDAF + AQA-10-Full | 51.0 | 61.2 | 48.4 | 58.7 | 1 | 4.6 |
| BiDAF + AQA-10-Full (extra budget) | 51.4 | 61.3 | 50.5 | 58.9 | 10 | 46.0 |
| BERT + AQA-10-Sub | 71.2 | 76.8 | 69.1 | 75.4 | 1 | 4.6 |

Table 2: Main result on the question-answering task (SearchQA dataset).

## 5 QUESTION-ANSWERING

On the question-answering task, we compare against the active question answering agent proposed by Buck et al. (2018b). The environment receives a question and returns an answer and a reward computed against a ground truth answer. We use either BiDAF (Seo et al., 2016) or BERT (Devlin et al., 2018) as a question-answering system. We use as a reward the token level F1 score on the answer (see Section 5.3). We follow Buck et al. (2018b) to train BiDAF and BERT. We emphasize that parameters are frozen when we train and evaluate the reformulation system. Training and evaluation are performed on the SearchQA dataset (Dunn et al., 2017). The data contains *Jeopardy!* clues as questions. Each clue has a correct answer and a list of 50 snippets from Google's top search results. The training, validation and test sets contain 99,820, 13,393 and 27,248 examples, respectively.

### 5.1 BASELINES AND BENCHMARKS

**BiDAF/BERT:** The original question is given to the question-answering system without any modification (see Figure 1-(a)).

**AQA:** The best model from Buck et al. (2018b). It consists of a reformulator and a selector. The reformulator is a subword-based seq2seq model that produces twenty reformulations of a question with beam search. Answers for the original question and its reformulations are obtained from BiDAF. These are given to the selector which then chooses one of the answers as final (see Figure 1-(b)). The reformulator is pretrained on zero-shot translation.

### 5.2 PROPOSED METHODS

**AQA-N-{Full, Sub}:** Similar to the RL-N-{Full, Sub} models, we use AQA reformulators as the sub-agents followed by an aggregator to create AQA-N-Full and AQA-N-Sub models, whose sub-agents are trained on the full and random partitions of the dataset, respectively. For the training and hyperparameter details, see Appendix B.2.

| Method | pCos ↓ | pBLEU ↓ | PINC ↑ | Length Std ↑ | F1 ↑ | Oracle ↑ |
|---|---|---|---|---|---|---|
| AQA | 66.4 | 45.7 | 58.7 | 3.8 | 47.7 | 56.0 |
| AQA-10-Full | 29.5 | 26.6 | 79.5 | 9.2 | 51.0 | 61.2 |
| AQA-10-Sub | **14.2** | **12.8** | **94.5** | **11.7** | **51.4** | **61.3** |

Table 3: Diversity scores of reformulations from different methods. For pBLEU and pCos, lower values mean higher diversity. Higher diversity scores are associated with higher F1/oracle scores.

## 5.3 EVALUATION METRICS

**F1:** We use the macro-averaged F1 score as the main metric. It measures the average bag of tokens overlap between the prediction and ground truth answer. We take the F1 over the ground truth answer for a given question and then average over all of the questions.

**Oracle:** Additionally, we present the oracle performances, which are from a perfect aggregator that predicts $s_j^R = 1$ for relevant answers and $s_j^R = 0$, otherwise.

## 5.4 RESULTS

Results are presented in Table 2. When using BiDAF as the Q&A system, our methods (AQA-10-{Full, Sub}) have both better F1 and oracle performances than single-agents AQA methods, while training in one-tenth of the time. Even when the ensemble method is given ten times more training time (AQA-10-Full, extra budget), our method performs better. We achieve state-of-the-art on SearchQA by a wide margin with BERT. Our reformulation strategy (BERT + AQA-10-Sub), however, could not improve upon this underlying Q&A system. We conjecture that, although there is room for improvement, as the oracle performance is 5-7% higher than BERT alone, the reformulations and answers do not contain enough information for the aggregator to discriminate good from bad answers. One possible way to fix this is to give the context of the answer to the aggregator, although in our experiments we could not find any successful way to use this extra information. We observe a drop in F1 of approximately 1% when the original query is removed from the pool of reformulations, which shows that the gains come mostly from the multiple reformulations and not from the aggregator falling back on selecting the original query.

## 5.5 QUERY DIVERSITY

In accordance with the 'mixture of experts' idea, we expected specialisation to be advantageous for agents and tried several meaningful clustering approaches (cf Appendix F). However, we surprisingly found that random clusterings were superior and query diversity being an important reason. We evaluate query diversity vs. performance using four metrics (see Appendix E): pCos, pBLEU, PINC, and Length Std. Table 3 shows that the multiple agents trained on partitions of the dataset (AQA-10-Sub) produce more diverse queries than a single agent with beam search (AQA) and multiple agents trained on the full training set (AQA-10-Full). This suggests that its higher performance can be partly attributed to the higher diversity of the learned policies.

## 6 CONCLUSION

We proposed a method to build a better query reformulation system by training multiple sub-agents on partitions of the data using reinforcement learning and an aggregator that learns to combine the answers of the multiple agents given a new query. We showed the effectiveness and efficiency of the proposed approach on the tasks of document retrieval and question answering. We also found that a first attempt based on semantic clustering did not produce good results, and that diversity was an important but hard to characterize reason for improved performance. One interesting orthogonal extension would be to introduce diversity on the beam search decoder (Vijayakumar et al., 2016; Li et al., 2016), thus shedding light on the question of whether the gains come from the increased capacity of the system due to the use of the multiple agents, the diversity of reformulations, or both. Furthermore, we found that reinforcement learning for the reformulation task is hard when the underlying system already performs extremely well on the task. This might be due to the tasks being too constrained (which makes it possible for machines to almost reach human performance), and requires further exploration.

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

## APPENDIX A DOCUMENT RETRIEVAL: RESULTS ON MORE METRICS

Following Dietz et al. (2017), we report the results on four standard TREC evaluation measures: R-Precision (R-Prec), Mean-average Precision (MAP), Reciprocal Rank (MRR), and Normalize Discounted Cumulative Gain (NDCG). We also include Recall@40 as this is the reward our agents are optimizing for. The results for TREC-CAR, Jeopardy, and MSA are in Tables 4, 5, 6, respectively.

## APPENDIX B HYPERPARAMETERS

### B.1 DOCUMENT RETRIEVAL TASK

SUB-AGENTS: We use mini-batches of size 256, ADAM (Kingma & Ba, 2014) as the optimizer, and learning rate of $10^{-4}$.

AGGREGATOR: The encoder $f_{q_0}$ is a word-level two-layer CNN with filter sizes of 9 and 3, respectively, and 128 and 256 kernels, respectively. $D = 512$. No dropout is used. ADAM is the optimizer with learning rate of $10^{-4}$ and mini-batch of size 64. It is trained for 100 epochs.

### B.2 QUESTION-ANSWERING TASK

SUB-AGENTS: We use mini-batches of size 64, SGD as the optimizer, and learning rate of $10^{-3}$.

AGGREGATOR: The encoder $f_{q_0}$ is a token-level, three-layer CNN with filter sizes of 3, and 128, 256, and 256 kernels, respectively. We train it for 100 epochs with mini-batches of size 64 with SGD and learning rate of $10^{-3}$.

## APPENDIX C AGGREGATOR ANALYSIS

### C.1 CONTRIBUTION OF THE AGGREGATOR VS. MULTIPLE REFORMULATORS

To isolate the contribution of the Aggregator from the gains brought by the multiple reformulators, we use the aggregator to re-rank the list of documents obtained with the rewrite from a single reformulator (RL-RNN Greedy + Aggregator). We also use beam search or sampling to produce $K$ rewrites from a single reformulator (RL-RNN $K$ Sampled/Beam + Aggregator). The $K$ lists of ranked documents returned by the environment are then merged into a single list and re-ranked by the Aggregator.

The results are shown in table 7. The higher performance obtained with ten rewrites produced by different reformulators (RL-10-Sub) when compared 20 sampled rewrites from a single agent (RL-RNN 20 Sampled + Aggregator) indicates that the gains the proposed method comes mostly from the pool of diverse reformulators, and not from the simple use of a re-ranking function (Aggregator).

### C.2 ABLATION STUDY

To validate the effectiveness of the proposed aggregation function, we conducted a comparison study on the TREC-CAR dataset. We present the results in Table 8. We notice that removing or changing the accumulated rank or relevance score functions results in a performance drop between 0.4-1.4% in MAP. The largest drop occurs when we remove the aggregated rank ($s_j = s_j^R$), suggesting that the rank of a document obtained from the reformulation phase is a helpful signal to the re-ranking phase.

Not reported in the table, we also experimented concatenating to the input vector $z_i$ (eq. 2) a vector to represent each sub-agent. These vectors were learned during training and allowed the aggregator to distinguish sub-agents. However, we did not notice any performance improvement.

| | R@40 | MAP | R-Prec | MRR | NDCG |
|---|---|---|---|---|---|
| BM25 | 27.5 | 11.3 | 9.8 | 21.0 | 27.4 |
| PRF | 28.6 | 11.7 | 10.1 | 21.7 | 27.2 |
| RM3 | 29.7 | 12.1 | 10.5 | 22.5 | 27.2 |
| RL-RNN | 31.6 | 12.7 | 10.9 | 23.6 | 28.3 |
| RL-10-Ensemble | 31.7 | 12.8 | 11.0 | 23.8 | 28.3 |
| RL-RNN Greedy + Aggregator | 32.0 | 12.8 | 11.1 | 23.2 | 29.0 |
| RL-RNN 20 Sampled + Aggregator | 32.5 | 12.0 | 11.2 | 23.1 | 29.3 |
| RL-RNN 20 Beam + Aggregator | 32.3 | 12.9 | 11.1 | 23.0 | 29.2 |
| RL-10-Full | 35.2 | 14.1 | 12.0 | 24.5 | 29.5 |
| RL-10-Bagging | 35.9 | 14.1 | 12.1 | 24.6 | 29.7 |
| RL-10-Sub | 36.7 | 14.2 | 12.1 | 25.0 | 29.5 |
| RL-10-Sub (Pretrained) | 36.9 | 14.4 | 12.3 | 25.0 | 29.8 |
| RL-10-Full (Extra Budget) | 37.7 | 14.8 | 12.5 | 25.3 | 29.1 |
| RL-10-Full (Ensemble of 10 Aggregators) | 39.5 | 17.7 | 13.5 | 26.3 | 30.8 |
| RM3 + BERT Aggregator | 50.9 | 35.5 | 32.1 | 51.1 | 45.2 |
| RL-10-Full + BERT Aggregator | 52.0 | 36.4 | 32.6 | 52.0 | 46.6 |
| Best System of TREC-CAR 2017 (MacAvaney et al., 2017) | - | 14.8 | 11.6 | 22.3 | 22.8 |

Table 4: Results on more metrics on the test set of the TREC-CAR dataset.

| | R@40 | MAP | R-Prec | MRR | NDCG |
|---|---|---|---|---|---|
| BM25 | 23.0 | 8.2 | 4.4 | 8.2 | 11.9 |
| PRF | 29.7 | 13.1 | 8.4 | 13.1 | 17.4 |
| RM3 | 30.5 | 13.5 | 8.7 | 13.5 | 17.9 |
| RL-RNN | 33.7 | 15.9 | 10.6 | 15.9 | 20.5 |
| RL-10-Ensemble | 35.2 | 17.0 | 11.4 | 17.0 | 21.8 |
| RL-RNN Greedy + Aggregator | 42.0 | 22.1 | 15.5 | 22.1 | 27.2 |
| RL-RNN 20 Sampled + Aggregator | 42.4 | 22.4 | 15.7 | 22.4 | 27.5 |
| RL-RNN 20 Beam + Aggregator | 42.3 | 22.3 | 15.6 | 22.3 | 27.3 |
| RL-10-Full | 52.1 | 29.3 | 21.1 | 29.3 | 35.1 |
| RL-10-Bagging | 52.5 | 29.6 | 21.4 | 29.6 | 35.4 |
| RL-10-Sub | 53.5 | 29.7 | 23.0 | 30.6 | 36.4 |
| RL-10-Sub (Pretrained) | 54.0 | 30.7 | 22.2 | 30.7 | 36.6 |
| RL-10-Full (Extra Budget) | 54.4 | 31.2 | 22.7 | 31.2 | 37.2 |
| RM3 + BERT Aggregator | 61.5 | 41.3 | 36.1 | 41.3 | 43.4 |
| RL-10-Full + BERT Aggregator | 62.7 | 42.5 | 37.0 | 42.5 | 44.8 |

Table 5: Results on more metrics on the test set of the Jeopardy dataset.

|  | R@40 | MAP | R-Prec | MRR | NDCG |
|---|---|---|---|---|---|
| BM25 | 12.7 | 3.1 | 6.0 | 15.4 | 9.1 |
| PRF | 13.2 | 3.4 | 6.4 | 16.2 | 9.7 |
| RM3 | 12.3 | 3.1 | 6.0 | 15.0 | 8.9 |
| RL-RNN | 15.1 | 4.1 | 7.3 | 18.8 | 11.2 |
| RL-10-Ensemble | 15.8 | 4.4 | 7.7 | 19.7 | 11.7 |
| RL-RNN Greedy + Aggregator | 16.1 | 4.5 | 7.8 | 20.1 | 12.0 |
| RL-RNN 20 Sampled + Aggregator | 16.4 | 4.6 | 7.9 | 20.5 | 12.2 |
| RL-RNN 20 Beam + Aggregator | 16.2 | 4.5 | 7.9 | 20.3 | 12.1 |
| RL-10-Full | 17.4 | 4.9 | 8.4 | 21.9 | 13.0 |
| RL-10-Bagging | 17.6 | 5.0 | 8.5 | 22.1 | 13.2 |
| RL-10-Sub | 18.9 | 5.5 | 9.2 | 23.9 | 14.2 |
| RL-10-Sub (Pretrained) | 19.1 | 5.4 | 9.1 | 24.0 | 14.2 |
| RL-10-Full (Extra Budget) | 19.2 | 5.6 | 9.3 | 24.3 | 14.4 |
| RM3 + BERT Aggregator | 22.7 | 6.6 | 8.9 | 33.0 | 16.2 |
| RL-10-Full + BERT Aggregator | 23.8 | 7.2 | 10.2 | 34.7 | 17.6 |

Table 6: Results on more metrics on the test set of the MSA dataset.

|  | TREC-CAR | Jeopardy | MSA |
|---|---|---|---|
| RL-RNN | 10.8 | 15.0 | 4.1 |
| RL-RNN Greedy + Aggregator | 10.9 | 21.2 | 4.5 |
| RL-RNN 20 Sampled + Aggregator | 11.1 | 21.5 | 4.6 |
| RL-RNN 20 Beam + Aggregator | 11.0 | 21.4 | 4.5 |
| RL-10-Sub | 12.3 | 29.7 | 5.5 |

Table 7: Multiple reformulators vs. aggregator contribution. Numbers are MAP scores on the dev set. Using a single reformulator with the aggregator (RL-RNN Greedy/Sampled/Beam + Aggregator) improves performance by a small margin over the single reformulator without the aggregator (RL-RNN). Using ten reformulators with the aggregator (RL-10-Sub) leads to better performance, thus indicating that the pool of diverse reformulators is responsible for most of the gains of the proposed method.

| Aggregator Function |  | MAP | Diff |
|---|---|---|---|
| $s_j = s_j^A s_j^R$ (proposed, Section 3.4) |  | 12.3 | - |
| $z_j = f_{\text{CNN}}(q_0) \| f_{\text{BOW}}(a_j)$ (eq. 2) |  | 11.9 | -0.4 |
| $s_j^A = \sum_{i=1}^{N} \mathbb{1}_{a_i = a_j}$ |  | 11.7 | -0.6 |
| $s_j = s_j^A$ |  | 11.1 | -1.2 |
| $s_j = s_j^R$ |  | 10.9 | -1.4 |

Table 8: Comparison of different aggregator functions on TREC-CAR. The reformulators are from RL-10-Sub.

| | SearchQA | | | | TREC-CAR | | | |
|---|---|---|---|---|---|---|---|---|
| | $E_i[e_i] \downarrow$ | $E_i[E_{j\neq i}[s_{ij}]] \uparrow$ | $E_i[V_{j\neq i}[s_{ij}]] \downarrow$ | F1↑ | $E_i[e_i] \downarrow$ | $E_i[E_{j\neq i}[s_{ij}]] \uparrow$ | $E_i[V_{j\neq i}[s_{ij}]] \downarrow$ | R@40↑ |
| Q | 9.9 | 52.0 | **1.1** | 53.3 | 15.3 | 50.4 | 5.9 | 50.0 |
| A | 22.0 | 50.1 | 3.9 | 51.4 | **1.3** | **57.0** | 0.3 | **56.9** |
| Q+A | **9.0** | 50.5 | 1.2 | **53.4** | 1.8 | 56.2 | 0.3 | 56.5 |
| Rand. | 9.5 | **53.8** | **1.1** | **53.4** | 1.9 | **57.0** | **0.2** | 57.1 |

Table 9: Partitioning strategies and the corresponding evaluation metrics. We notice that the random strategy generally results in the best quality sub-agents, leading to the best scores on both of the tasks.

## APPENDIX D    TRAINING STABILITY OF SINGLE VS. MULTI-AGENT

Reinforcement learning algorithms that use non-linear function approximators, such as neural networks, are known to be unstable (Tsitsiklis & Van Roy, 1996; Fairbank & Alonso, 2011; Pirotta et al., 2013; Mnih et al., 2015). Ensemble methods are known to reduce this variance (Freund, 1995; Breiman, 1996a;b). Since the proposed method can be viewed as an ensemble, we compare the AQA-10-Sub's F1 variance against a single agent (AQA) on ten runs. Our method has a much smaller variance: 0.20 vs. 1.07. We emphasize that it also has a higher performance than the AQA-10-Ensemble.

We argue that the higher stability is due to the use of multiple agents. Answers from agents that diverged during training can be discarded by the aggregator. In the single-agent case, answers come from only one, possibly bad, policy.

## APPENDIX E    DIVERSITY METRICS

Here we define the metrics used in query diversity analysis (Sec. 5.5):

PCOS:    Mean pair-wise cosine distance: $\frac{1}{N} \sum_{n=1}^{N} \frac{1}{|Q^n|} \sum_{q,q' \in Q^n} \cos(\#q, \#q')$, where $Q^n$ is a set of reformulated queries for the $n$-th original query in the development set and $\#q$ is the token count vector of q.

PBLEU: Mean pair-wise sentence-level BLEU (Chen & Cherry, 2014): $\frac{1}{N} \sum_{n=1}^{N} \frac{1}{|Q^n|} \sum_{q,q' \in Q^n} \text{BLEU}(q, q')$.

PINC : Mean pair-wise paraphrase in k-gram changes (Chen & Dolan, 2011): $\frac{1}{N} \sum_{n=1}^{N} \frac{1}{|Q^n|} \sum_{q,q' \in Q^n} \frac{1}{K} \sum_{k=1}^{K} 1 - \frac{|\text{k-gram}_q \cap \text{k-gram}_{q'}|}{|\text{k-gram}_{q'}|}$, where $K$ is the maximum number of k-grams considered (we use $K = 4$).

LENGTH STD:    Standard deviation of the reformulation lengths: $\frac{1}{N} \sum_{n=1}^{N} \text{std}(\{|q_i^n|\}_{i=1}^{|Q|})$

## APPENDIX F    ON DATA PARTITIONING

Throughout this paper, we used sub-agents trained on random partitions of the dataset. We now investigate how different data partitioning strategies affect final performance of the system. Specifically, we compare the random split against a mini-batch K-means clustering algorithm (Sculley, 2010).

**Balanced K-means Clustering**    For K-means, we experimented with three types of features: average question embedding (Q), average answer embedding (A), and the concatenation of these two (Q+A). The word embeddings were obtained from Mikolov et al. (2013).

The clusters returned by the K-means can be highly unbalanced. This is undesirable since some sub-agents might end up being trained with too few examples and thus may have a worse generalization performance than the others. To address this problem, we use a greedy cluster balancing algorithm as a post-processing step (see Algorithm 1 for the pseudocode).

---

**Algorithm 1** Cluster Balancing
---
1: Given: desired cluster size $M$, and a set of clusters $C$, each containing a set of items.
2: sort $C$ by descending order of sizes
3: $C_{\text{remaining}} \leftarrow \text{shallow\_copy}(C)$
4: **for** c in C **do**
5:     remove $c$ from $C_{\text{remaining}}$
6:     **while** c.size $< M$ **do**
7:         item $\leftarrow$ randomly select an item from c
8:         move item to the closest cluster in $C_{\text{remaining}}$
9:         sort $C_{\text{remaining}}$ by descending order of sizes
10:     **end while**
11: **end for**
12: **return** $C$
---

**Evaluation Metric** In order to gain insight into the effect of a partitioning strategy, we first define three evaluation metrics. Let $\pi_i$ be the $i$-th sub-agent trained on the $i$-th partition out of $K$ partitions obtained from clustering. We further use $s_{ij}$ to denote the score, either F-1 in the case of question answering or R@40 for document retrieval, obtained by the $i$-th sub-agent $\pi_i$ on the $j$-th partition.

**Out-of-partition score** computes the generalization capability of the sub-agents outside the partitions on which they were trained:

$$E_i[E_{j \neq i}[s_{ij}]] = \frac{1}{N} \sum_{i=1}^{N} \frac{1}{K-1} \sum_{j \neq i} s_{ij}.$$

This score reflects the general quality of the sub-agents. **Out-of-partition variance** computes how much each sub-agent's performance on the partitions, on which it was not trained, varies:

$$E_i[V_{j \neq i}[s_{ij}]] = \frac{1}{N} \sum_{i=1}^{N} \frac{1}{K-2} \sum_{j \neq i} (s_{ij} - E_{i \neq j}[s_{ij}])^2. \tag{6}$$

It indicates the general stability of the sub-agents. If it is high, it means that the sub-agent must be carefully combined in order for the overall performance to be high. **Out-of-partition error** computes the generalization gap between the partition on which the sub-agent was trained and the other partitions:

$$E_i[e_i] = \frac{1}{N} \sum_{i=1}^{N} (s_i j - E_{j \neq i}[s_{ij}]).$$

This error must be low, and otherwise, would indicate that each sub-agent has overfit the particular partition, implying the worse generalization.

**Result** We present the results in Table 9. Although we could obtain a good result with the clustering-based strategy, we notice that this strategy is highly sensitive to the choice of features. Q+A is optimal for SearchQA, while A is for TREC-CAR. On the other hand, the random strategy performs stably across both of the tasks, making it a preferred strategy. Based on comparing Q and Q+A for SearchQA, we conjecture that it is important to have sub-agents that are not specialized too much to their own partitions for the proposed approach to work well. Furthermore, we see that the absolute performance of the sub-agents alone is not the best proxy for the final performance, based on TREC-CAR.

## APPENDIX G   REFORMULATION EXAMPLES

Table 10 shows four reformulation examples by various methods. The proposed method (AQA-10-Sub) performs better in the first and second examples than the other methods. Note that, despite the large diversity of reformulations, BiDAF still returns the correct answer.

In the third example, the proposed method fails to produce the right answer whereas the other methods perform well. In the fourth example, despite the correct answer is in the set of returned answers, the aggregator fails to set a high score for it.

| Method | Query | Reference / Answer from BiDAF (F1) |
|---|---|---|
| Jeopardy! | The name of this drink that can be blended or on the rocks means "daisy" in Spanish | |
| SearchQA | name drink blended rocks means daisy spanish | margarita |
| AQA | What name drink blended rocks mean daisy spanish? | margarita tequila daisy (0.33) |
| | **What rock drink name means daisy spanish?** | **margarita tequila daisy mentioned (0.20)** |
| | What name drink blended rocks means daisy spanish? | margarita tequila daisy mentioned (0.20) |
| | What rock drinks name means daisy spanish? | margarita tequila daisy mentioned (0.20) |
| | What name drink blended rock means daisy spanish? | margarita tequila daisy mentioned (0.20) |
| AQA-10-Full | What is drink name name drink daisy daisy? me | margarita eater jun (0.33) |
| | What name is drink spanish? | margarita eater jun (0.33) |
| | **What is daisy blender rock daisy spanish?? daisy spanish?** | **cocktail daisy margarita spanish (0.26)** |
| | rock name name | cocktail daisy margarita spanish (0.25) |
| | What name drink blended st st st st st ship ship | cocktail daisy margarita spanish (0.26) |
| AQA-10-Sub | Where is name drink?? | margarita (1.0) |
| | **What is drink blended rock?** | **margarita (1.0)** |
| | rock definition name | margarita (1.0) |
| | What is name drink blended rock daisy spanish 16 daisy spanish? | margarita similarity (0.5) |
| | Nam Nam Nam Nam Nam Nam Nam drink | tequila (0.0) |
| Jeopardy! | A graduate of Howard University, she won the Nobel Prize for literature in 1993 | |
| SearchQA | graduate howard university , nobel prize literature 1993 | toni morrison |
| AQA | Nobel university of howard university? | toni morrison american novelist (0.5) |
| | Nobel university of howard university in 1993? | toni morrison american novelist (0.5) |
| | **Nobel graduate literature in 1993?** | **toni morrison american novelist (0.5)** |
| | Nobel university graduate howard university 1993? | princeton (0.0) |
| | Nobel university for howard university? | columbia (0.0) |
| AQA-10-Full | Another university start howard university starther | toni morrison american novelist (0.5) |
| | **university howard car?** | **toni morrison american novelist (0.5)** |
| | What is howard graduate nobel? | toni morrison american novelist (0.5) |
| | What is howard howard university? | toni morrison american novelist (0.5) |
| | Where graduated howard university noble literature literature? | american novelist morrison (0.16) |
| AQA-10-Sub | **Where is howard university??** | **toni morrison (1.0)** |
| | The nobel university? | toni morrison (1.0) |
| | What name howard howard university? | toni morrison (1.0) |
| | This howard? | toni morrison american novelist (0.5) |
| | 1993? | howard cornell universities (0.0) |
| Jeopardy! | For Bill Gates, it computes to own 2 models, the 959 and the 911, from this manufacturer | |
| SearchQA | bill gates , computes 2 models , 959 911 , manufacturer | porsche |
| AQA | **Bill gates iin computes older models?** | **porshe (1.0)** |
| | Bill gates in compute gates how old are they? | porshe (1.0) |
| | Bill gates bill gates computes mod? | porshe (1.0) |
| | Bill gates computes 2 models pics of 959? | porshe (1.0) |
| | Bill gates in compute gates how old is it? | porshe (1.0) |
| AQA-10-Full | Another model start bill bette | porshe (1.0) |
| | What is an bill gates 100 car? | porshe (1.0) |
| | What is bill bill bill bill gates computes? | porshe (1.0) |
| | **What is manufacturer?** | **porshe (1.0)** |
| | bill bill gats sa computes 2 bill gats? | porshe (1.0) |
| AQA-10-Sub | Where is bill gates manufacturer? | bill gates (0.0) |
| | **A bill gates?** | **bill gates (0.0)** |
| | The model? | bill gates (0.0) |
| | What is bill gates model? | sports car (0.0) |
| | What model bill gates 9 58 model 9 gates? | sports car (0.0) |
| Jeopardy! | The first written mention of this capital's name was in a 1459 document of Vlad the Impaler | |
| SearchQA | first written mention capital 's name 1459 document vlad impaler | bucharest |
| AQA | First film was written by 1459 vlad impaler? | bucharest castle (0.5) |
| | First film was written by 1459 vlad impalter? | bucharest castle (0.5) |
| | First film was written by 1459 vlad impal? | bucharest castle (0.5) |
| | First film was written by 1459 vlad impalot? | bucharest castle (0.5) |
| | **First film was written in 1459?** | **bucharest national capital (0.33)** |
| AQA-10-Full | What is capital vlad impaler? | bucharest (1.0) |
| | First referred capital vlad impaler impaler? | bucharest (1.0) |
| | capital | romania 's largest city capital (0.0) |
| | Another name start capital | romania 's largest city capital (0.0) |
| | **capital capital vlad car capital car capital?** | **romania 's largest city capital (0.0)** |
| AQA-10-Sub | Where is vla capital capital vlad impalers? | bucharest (1.0) |
| | What capital vlad capital document document impaler? | bucharest (1.0) |
| | **Another capital give capital capital** | **bulgaria , hungary , romania (0.0)** |
| | capital? | bulgaria , hungary , romania (0.0) |
| | The name capital name? | hungary (0.0) |

Table 10: Examples for the qualitative analysis on SearchQA. In **bold** are the reformulations and answers that had the highest scores predicted by the aggregator. We only show the top-5 reformulations of each method. For a detailed analysis of the language learned by the reformulator agents, see Buck et al. (2018a).

