# OpenReview forum: "Multi-agent query reformulation: Challenges and the role of diversity"
_ICLR.cc/2019/Workshop/drlStructPred — drlStructPred 2019_

### Official Review · AnonReviewer1 · 2019-03-31
**Clear paper but lack of novelty and marginal improvement**

**Rating:** 3
**Confidence:** 2

**Review:**


This work applies the mixture-of-experts approach to the query reformulation task, where ones goals is to reformulate queries in order to optimize the quality of results returned by an underlying retrieval system. The authors propose to train multiple reformulators (sub-agents) each on a subset of the training dataset, leading to a diverse set of reformulators. Given a new query, each reformulator can thus propose different query reformulations, which are then fed (in addition to the initial query) to an underlying search engine. For each query, the search engine outputs a recommendation. All recommendations are sent to an aggregator (meta-agent), which decides which one to pick. The authors propose an architecture and loss function to train the aggregator.

As pointed out, the idea of using a mixture-of-experts trained on different subsets of data is old and well-known. Results showing that ensembles of such sub-agents produce diverse reformulations are therefore not surprising.

Impact of aggregator:
Appendix C shows the benefits of the ensemble of sub-agents independently from the effect of using an aggregator (Fig.1c) instead of a selector (Fig.1b). However, it seems like anything combined with BERT (alone or as an aggregator) increases the performance so much, therefore making the benefits of using diverse sub-agents appear as marginal. It is not clear that relying on diverse ensembles would even bring any improvement given a strong aggregator.

Diversity of sub-agents:
How does the size of sub-datasets used by sub-agents affect the performance and diversity? Is there any overlap between sub-datasets used by sub-agents? If not, it means that the number of sub-agents affects the size of dataset available to each sub-agent. How would that affect the performance in return? If sub-datasets are allowed to overlap, should we expect a tradeoff between large sub-datasets (increasing overlap, therefore decreasing diversity) and small sub-datasets (more diversity, maybe worse generalization)?

Overall, although the approach lacks novelty and results are either expected or not impressive, the paper is well written, easy to follow, and presents a decent empirical evaluation. I therefore think that this paper is suitable for a workshop.

Minor questions/comments:
* [Sec.2] Small communication overhead: A downside of the approach of Shazeer et al. (2017) is that it requires that "output vectors of experts are exchanged between machines." The authors claim that their method instead "requires only scalars (rewards) and short strings (original query, reformulations, and answers) to be exchanged." How are reformulations and answers different from the output vectors?
* [Sec.3.3] Training of sub-agents: "At training time, instead of using beam search, we sample reformulations." From which distribution are reformulation sampled?
* [Sec.3.4] Training of aggregator: Why is the loss only using the relevance score (see Eq.3)?
* [Sec.4.3] Reward: Any insights regarding why agent trained to optimize Recall@40 perform better than those trained to optimize MAP, even though MAP is used in evaluation?
* [Sec.4.5] Proposed model RL-N-Full: What is the "best reformulations of all the agents"?
* [Sec.4.6, 2nd paragraph] "...an RL-10-Full with an ensemble of 10 aggregators yields a relative performance...": Should "aggregators" be replaced by "reformulators"?
* [Tables 1 and 2] Consistency of training time in sub-agents+BERT: Why does RL-10-Sub + BERT Aggregator require 10 training days (Tab.1) while BERT + AQA-10-Sub require 1 training day (Tab.2)?

---

### Official Review · AnonReviewer4 · 2019-04-05
**Hierarchical RL applied to document retrieval and QA. Interesting but somewhat disappointing results.**

**Rating:** 3
**Confidence:** 2

**Review:**

This paper instantiates in the context of query reformulation an approach to structure prediction (seq2seq) that is inspired by hierarchical reinforcement learning methods proposed in the early nineties by Singh, Lin, Dietterich, and Hinton.

This is an experimental paper with well executed experimental design and a broad range of comparative results for doc retrieval and question answering. Although experimentally solid, the paper is light in terms of insights.

On the positive side, the approach is easily parallelizable and achieves better generalization performance than a model average ensemble. On the negative side, the approach does not lead to significant improvements.

---

### Official Review · AnonReviewer2 · 2019-04-05
**Simple idea, strong empirical results, but some results are surprising**

**Rating:** 3
**Confidence:** 2

**Review:**

The authors propose a multi-agent approach for query reformulation. A set of sub-agents is trained on disjoint splits of the training data, and a meta-agent (the aggregator) is trained on the whole dataset and learns to combine outputs from the sub-agents.  The proposed approach is evaluated on two tasks: document retrieval and question answering. For document retrieval, recall@40 is used as the reward signal. The token level F1 score on the answer is used as the reward signal for question answering.

Strength:
========

- The proposed approach is simple, easy to parallelize, and hence computational efficiency can be achieved.
- Strong empirical results, although no significant improvement was observed when improving over a BERT model on the question answering task.
- Paper is clear and easy to follow.

Weakness:
=========
1) Some results are surprising and counterintuitive, and hence further analysis is needed for a better understanding:

a) What was the size of the random partitions used for training the sub-agents? I expect the dataset has been split equally between the agents in the RL-N-Sub setting, have the authors experimented with other partition sizes in addition to training the sub-agents on the full dataset?
b) The results described in section 5.5 are surprising, perhaps the reason for this results is limited to K-means clustering which is prone to fail when the clusters have non-spherical structures. Have the authors experimented with other density based clustering algorithms (e.g. dbscan)?
c) Not being able to significantly improve on BERT in the question answering task is concerning.

2) Originality is limited, the main difference between the proposed approach and previous work seems to be the training on disjoint splits for the sub-agents, and then training an aggregator to combine the results.

Clarity:
======

The paper is clear and well written.

---

### Official Review · AnonReviewer5 · 2019-04-05
**Elegant idea, preliminary results**

**Rating:** 3
**Confidence:** 2

**Review:**

This paper explores whether decomposing an agent-oriented structure prediction task into sub-agents and an aggregator improves performance on query reformulation tasks.  The paper considers both decomposing on random sub-sets of the data (bagging), and partitioning into semantically similar classes.  The results show that decomposing into sub-agents improves performance.

The method is applied to 2 tasks - document retrieval (on 3 datasets), and question answering.  One of the main results is that the same level of performance can be achieved with much less compute.

The basic idea is elegant, but I have 3 concerns. First, I found that some aspects were really unclear.  Second, it wasn’t clear but I inferred that the results were created from a single run, which limits how much can be concluded.  Finally, it wasn’t clear whether the method would generalize to other tasks.

* Clarity.  I was unclear on the following:

- Fig 1: why is there 1 search box in Fig 1b, but N+1 search boxes in Fig 1c?  If the idea is that the search in Fig 1b is doing a single search by simultaneously receiving all queries, that wasn’t at all clear.

- Sec 3.4, “the aggregator receives as input q_0…”: q_0 is not an input into the aggregator in Fig 1.  Should q_0 be a_0?
- Sec 4.3, “other metrics … resulted in similar or slightly worse performance than Recall@40.”  Does this mean that the proposed method is strongest only when evaluated under certain metrics?
- Table 1: I was unclear on how RL-10-Ensemble and RL-10-Full differ, since the descriptions in the text (Sec 4.4 and 4.5) are nearly identical.
* Experimental procedure.  Were the results the product of a single run, or the results of averaging many runs?  I might have missed it but I inferred they were from a single run.  As a result, readers won’t know what the variance of the methods are, and whether the relative orderings of the methods are reliable.

* Generality.  Query re-writing is a very specific task, and both experiments here relied on an external search engine.  I’m worried that the method may be “fit” to specific properties of the task — for example, in the current setting, all actions can effectively be executed to produce search results; in other settings, this isn’t possible.  Moreover, because of the search engine, producing a diversity of responses seems to be important; in other settings, having a diverse set of outputs may be less important.  The results would be more impactful if other types of tasks were considered — even more text generation tasks, like dialog response generation, paraphrasing, translation, etc.

---

### Decision · Program_Chairs · 2019-04-09
**Acceptance Decision**

Accept